# Significance of Group Composition for the Welfare of Pastured Horses

**DOI:** 10.3390/ani9010014

**Published:** 2019-01-05

**Authors:** Hrefna Sigurjónsdóttir, Hans Haraldsson

**Affiliations:** 1Faculty of Subject Teacher Education, School of Education, University of Iceland, Stakkahlíð, R105 Reykjavík, Iceland; 2Educational Research Institute, School of Education, University of Iceland, Stakkahlíð, R105 Reykjavík, Iceland; haha@hi.is

**Keywords:** horse welfare, aggression, allogrooming, pastured horses, Icelandic horse

## Abstract

**Simple Summary:**

Because of their social nature, horses need to have plenty of opportunities to interact with others to establish bonds and learn from their elders. Comparison of social behaviour of 426 horses in 20 groups of Icelandic horses in pastures, showed that aggression was lowest where the group composition was like the natural system, i.e., with a stallion, mares and their young foals. In groups without a stallion, the presence of foals is also associated with low aggression. Stability of the group with respect to group composition is of great importance; the horses are less aggressive in the more stable groups. The highest aggression was found in groups of unfamiliar yearlings. The horses allogroomed more in groups with relatively more young horses, which suggests they are forming bonds. Later, they groom less but prefer certain individuals. Horse owners should all be aware of the importance of planning the composition of horse groups and to keep the membership as stable as possible in order to ensure good welfare.

**Abstract:**

We explore how herd composition and management factors correlate with frequencies of social interactions in horse groups. Since the welfare of horses correlates with low aggression levels and social contact opportunities, information of this kind is important. The data are a collection of records of social interactions of 426 Icelandic horses in 20 groups of at least eight horses. The complexities and limitations of the data prohibit useful statistical modelling so the results are presented descriptively. Interesting and informative patterns emerge which can be of use both in management and in future studies. Of special interest are the low levels of agonistic behaviours in breeding groups where one stallion was present. The horses were less agonistic when in groups with young foals and where group membership was stable. Unfamiliar yearlings in peer groups were especially aggressive. Allogrooming was most frequent in groups with relatively more young horses and in unstable and small groups. Interestingly, the horses allogroomed more if they had few preferred allogrooming partners. The findings show that composition (age/sex) and stability of groups are of great importance with respect to aggression levels and opportunities for establishing bonds.

## 1. Introduction

Consideration of the welfare of farm animals has been focused on physical well-being for most of the last century. In recent years, the need to address social needs as well has received increased attention [1,2]. By nature, horses are highly social animals, depending on the group for survival. Social bonds are likely to play an important role for social cohesion in group living domestic horses [3]. Management practices that minimizes aggression and give the horses ample opportunities to take part in affiliative interactions are to be recommended. In this study, data from several independent studies of social interactions among pastured horses in Iceland are used in order to identify factors that are associated with low levels of agonistic behaviour but enhance opportunities for behaviour that strengthens bonds among horses, such as mutual grooming.

Horses are highly social and as recent research has shown [4,5] their cognition level and learning abilities are well advanced. Unusually long-lasting bonds between individuals in a family band [6] may have contributed to the evolution of complex cognitive skills. Affiliation bonds between individuals are inferred from the choice of preferred allogrooming partners, play partners, a tendency to stay with certain individuals when resting, grazing or traveling or from affiliative approaches [7,8,9,10]. It has been argued that close bonds such as these are of the same nature as between friends among humans and should, therefore, be called friendships [11,12]. Horses have many ways of recognizing their group members [13,14], for instance by employing long-term memories of previous interactions [15] and using their ability to discriminate between individuals [16,17] and recognize social status of familiar horses relative to their own [18]. It is fair to conclude that horses can predict the outcome of encounters with familiar individuals and find ways of reducing tensions [19,20,21]. Studying their social interactions and comparing the nature and frequency of such interactions in different social environments can improve understanding of the cognitive abilities of horses. 

Group stability with respect to membership is high in most semi-feral groups [22]. This allows both stable dominance relationships and stable networks of friendships, the latter most often established between two horses of similar age and the same sex [6,9,23]. With frequent changes in group composition, levels of interactions are higher, especially agonistic interactions between resident horses and newcomers [24,25]. Such a lack of group stability is considered to be one of the risk factors for potential injuries [25].

Feral horses show low levels of aggressive interactions [6,21,26]. The same is true for groups of high stability kept under semi-natural conditions [23,25]. Fureix et al. proposed that the reasons for low aggression frequencies in such groups are the following: evolved ritualized communication signals, stable group composition, stable dominance hierarchies and opportunities for young animals to learn social skills [25].

Much less is known about frequencies of allogrooming than agonistic behaviours. The frequency of mutual grooming is very variable and may depend on a variety of factors, such as weather, parasites and social factors [21]. Different methodologies and limited information in published papers often make direct comparison of data impossible [9]. In addition to serving the role of forming social bonds [11,27], allogrooming following agonistic events is likely to reduce tension [28]. Feh and Maziéres [7] and Keiper [29] suggest that allogrooming lowers both heart rate and blood pressure and reduces social stress in horses. 

In contrast to the feral/natural situations where one stallion (occasionally two or three) keeps breeding mares with foals and their sexually immature offspring in a band (on average four to six horses [3]), groups of domestic horses vary greatly with respect to composition and environment. Typically, groups of horses are kept in pastures or in open stables or are let outdoors into paddocks for part of the day. These groups can be composed of peers only or mixed with respect to sex and age [24,30]. The composition of a group is known to influence the type and frequencies of social interactions [24,31,32]. Other factors are also known to be influential, such as the stability of the group, access to resources, density and the presence of a stallion [13,23]. However, as Hartmann et al. have pointed out, research on the effects of group size, group density and group composition on the social behaviour of horses is very limited [13]. In the present study, our main aim is to investigate how group composition and other variables (see below) influence the frequencies of both agonistic and affiliative behaviour in horses. The data are based on independent studies of 20 groups of Icelandic horses, carried out over a period spanning 15 years and involving a total of 426 horses, all kept on pastures of similar vegetation. The same methodology was applied in all studies. All the members of the studied groups belong to the Icelandic breed, which is the only breed found in the country, making the question about possible breed characteristics irrelevant. Here we use the individual data to show the overall frequency distributions of different types of behaviour, compare the sexes and estimate the effect of age. The group factors considered are group size, density of horses in the pasture, sex ratio, proportion of adults, numbers of young foals present, median number of friends, group stability and the presence of stallions. The environmental factors considered are season and whether or not hay was provided 

Results from the majority of the studies have been published previously [9,23,33,34,35,36,37,38]. The first published study in 2003, which featured a mixed, stable group without a stallion, indicated that the horses were more socially active compared to studies conducted in other countries of groups with a stallion [9]. Later, a comparison between four mixed groups in Iceland without a stallion and six groups with a stallion confirmed this; in groups where a stallion was present, the horses showed less agonistic behaviour [23]. Stallions play an important role by herding their mares and interacting with youngsters. Their behaviour might, therefore, suppress the mares’ social activities. However, the same study showed that the stallions were not dominant over the mares and they did not intervene actively during ongoing interactions between the mares [23,34]. 

By combining the data from the 20 groups, we intend to strengthen previous analyses of relations between social behaviour and group features. Furthermore, the combined data may reveal unknown influences of group features on social behaviour. It is hoped that the results will encourage horse owners to take up management practices that keep levels of aggression at a minimum level and increase opportunities for affiliative interactions, thus improving horses’ welfare.

## 2. Materials and Methods

### 2.1. Study Groups 

The data are drawn from independent studies of social interactions of individuals in 20 groups of Icelandic horses that were undertaken in the period 1997–2012 (Table 1). The total number of horses studied was 458 between 9 months and 26 years of age. Some individuals (32) were present in two groups, but in the analysis they were randomly deleted from one of the two. The group characteristics (e.g., proportion of males) were, however, based on the total number of horses in each group. The groups were located at farms in different parts of Iceland (Table 1). The group composition was variable. Three groups (A, B, C) included only sub-adults (1–3 years old), two groups (K, M) were composed of adults only (4 years and older) and six groups (P, Q, R, S, T, U) were breeding groups with one stallion each, where young foals (less than one month old) accompanied their dams. In four (R, S, T, U) of those groups some sub-adults were present (9 months to 3 years old). Two groups (D, F) which did not have a stallion included dams and young foals in addition to sub-adults of both sexes and adult geldings. In three cases, the same pasture was used in different years for different groups (applies to groups D + E, J + L, and K + M, respectively). Other places/pastures were only used once. The four breeding groups at Sel (Table 1) shared the same pasture (215 ha). Although the stallions kept their bands within their home range most of the time [34], all groups were seen to travel out of their home ranges at times. Therefore, the pasture size for all was set at 215 ha. These groups represent most types of groups one can expect to come across in Iceland. This allows us to compare frequency distributions in some basic classes like sex and age, as well as looking for patterns that might be influenced by group composition.

### 2.2. Management

All the groups were kept in spacious lowland pastures (median density 0.78 ha/horse, interquartile range (IQR) 1.32) with mixed vegetation of grasses, sedges, moss, perennial flower species, heath vegetation, small stony hills, running creeks and some wetland (no trees or bushes). No fertilizers are applied in these pastures and the grass is not cut. Most of the pastures are characterized by small tussocks dominated by grasses or sedges. The groups (J, K, L, M, O) that were observed in the winter months (January, February, March and April) were all provided with supplementary hay and also groups D and F in spring (Table 1). In the wintertime, the pastures are often free of snow and ice, in which case the horses had the same access to the grazing area as during other times of the year. In all cases, the resource availability (opportunity to graze, quantity of grass and other vegetation, access to hay and water) was such that the welfare of the horses was ensured.

In Iceland, training generally starts when horses are 4–5 years old. Until that time, they are kept outside throughout the year in groups. The horses in these studies had all been handled, some extensively but some very little. Some were stabled at six months of age for halter training but then released. Colts are generally castrated when 11 months old, and all the sub-adults were treated for internal parasites once a year. Most of the geldings in the groups were or had been used for riding and many of the mares too.

### 2.3. Recording of Social Interactions

All obvious social interactions of an agonistic (aggressive, submissive) nature and one type of affiliative behaviour, i.e., mutual grooming (allogrooming), were recorded, except those that involved the young foals. The method used was “All occurrence of some behaviours” [39], where the type of interaction and the identity of the horses involved in each case were recorded. The definitions of the types of behaviour (Table 2) were the same as given in McDonnell [8]) and used in Granquist et al. [23]. As argued by van Dierendonck et al. [40], it is controversial to include defensive acts when assessing the aggressive nature of individuals. Therefore, the behaviour types “threat to kick” and “kick” were not included as a part of aggressive encounters in the studies reported here. In the most extensive studies in this research [9,35], comparison of these two behaviour types with the other aggressive acts shows that threat with ears is by far the most dominant aggressive acts, especially amongst the adult horses (unpublished results). On average threat to kick and threat to bite combined were 20% of the other aggressive types. This has to be remembered when aggression rates are compared to studies where these behaviour types are included [25].

Horses were recognized by their characteristics, e.g., colour and colour patterns, size, type of mane, etc. Very similar horses in the same group were further marked by coloured tape in the mane or tail. The marking did not influence the behaviour of the horses. Care was taken not to interact with the horses in the field. If a horse came too close to an observer, the observer would move away slowly. Binoculars were used to observe the horses if they were some distance away. Data were recorded with a pencil by hand, in notebooks or in a handheld computer (Psion), and transferred into computer spreadsheets. Observation times varied between groups (range 40–848 h) (Table 1), the median duration being 78 h. As the interactions in question are easily observed and observers could see all horses at all times during observation periods, we can assume that all interactions were recorded. 

Inter-observer reliability tests were carried out in the first two most extensive studies (D and F groups) and the persons involved trained until the tests showed more than 90% agreement [9,35,36]. In later studies, the first author of this paper taught and trained the participants and took an active part in data collection at the start of each study. Thus, standardization in methodology was ensured. In total, 15 colleagues, students and assistants were involved in collecting the observational data (see Acknowledgements). Results from many of the studies (groups A, B, D, F, N, O, P, Q, R, S, T, U) have been published (Table 1) [9,23,35,36,37,38].

### 2.4. Statistics

The dataset consists of observations of individual horses clustered in groups. The groups were selected as examples of groups of certain types (breeding groups with a stallion, mixed groups without a stallion, sub-adult groups) and under certain conditions (e.g., season) for various prior research projects (see Table 1). The response variables are measures of the behaviour types aggression, submission and allogrooming. For data at the individual level, as well as the effects of season and group type, the bivariate relationship between behaviour frequencies, i.e., per hour per horse and individual characteristics is presented. For group level data, the relationship between median behaviour frequencies in the group and group variables (Table 3) is presented. As the distributions of behaviour frequencies are highly positively skewed, we always present the median as a measure of central tendency and all correlation coefficients presented are Spearman rank-order correlation coefficients. All statistical analyses were conducted using R 3.4.4 [41].

Graphic examination of the data shows strong indications of interactions between group- and individual-level variables. The default option for analysing data of this kind would normally be multilevel regression analysis but the non-random selection of groups and the relatively small number of groups observed renders that approach unfeasible. Therefore, we opt for an extensive descriptive treatment of the data over modelling. We present the bivariate relationships of individual-level variables and the behaviour frequencies of interest, ignoring the clustering. For the effects of group-level variables, we provide graphic presentation of the data indicating the most likely confounds of group type and season with colours, point shapes or by splitting plots. As the number of figures is large, we have created a scheme of colours and shapes which we follow consistently across figures to aid interpretation (e.g., red and blue for the sexes).

In the boxplots, the box represents the range from the 25th (1st quartile) to the 75th percentile (3rd quartile), with a bar in between representing the median. The difference between the 3rd and 1st quartile is the interquartile range (IQR). The whiskers represent the range of the data above the 3rd and below the 1st quartile up to a distance of 1.5 IQRs. Data points further away from the quartiles (outliers) are represented by points. 

Aggression data from one extreme outlier with an aggression frequency of 8.29 was removed from the boxplots. This was done in order to limit the span of the *y*-axis and make the other information in the plot (median, IQR, location of other outliers) more legible.

## 3. Result

The data from the 426 horses showed that the frequency distribution of the three different types of social interactions (per horse/hour), (aggression, submission, and allogrooming) were highly skewed (Figure 1).

High social activity is mostly limited to a few individuals while the majority of horses interacted at lower levels as reflected by the medians and their IQR. Out of 58 outliers in aggression frequency (i.e., individuals represented with points in Figure 1) six individuals are also outliers in submission frequency and 14 are also outliers in allogrooming frequency. Out of 36 outliers in submission frequency, eight individuals are also outliers in allogrooming frequency. Thus it is safe to say that the most aggressive individuals are generally not the same as those that are most active in submission and allogrooming.

### 3.1. Behaviour of Individuals 

The data on individual horses are derived from independent studies which involve varying group compositions (Table 1). Comparison of sexes suggests that agonistic encounters (aggression and submission) were more frequent in male horses while allogroming frequencies were similar in the sexes (Figure 2).

There was little difference in median aggression among age groups except for the oldest horses which were more aggressive than younger horses. The submission frequency was high among the oldest horses and the youngest horses. Allogrooming frequencies were higher among the youngest horses (Figure 3).

There were clear differences in frequencies of agonistic interactions (aggression and submission) between certain types of groups, the most pronounced contrast being the low frequencies of aggression and submission among the horses in the groups where a stallion was present (groups P, Q, R, S, T and U) and the high frequencies in the two yearling groups, A and B (Figure 4).

Figure 5 shows median allogrooming frequencies for the groups. The difference between the groups is less than for aggression and submission. The yearlings in the two small sub-adult groups, A (geldings) and B (fillies), were more socially active than other horses. 

Frequencies of agonistic interactions clearly varied with season, being lowest in the spring (May–June) and highest in the winter time (winter groups were K, L, M and O). Allogrooming frequencies were similar in all seasons (Figure 6).

In the wintertime, the horses were clearly most aggressive and submissive (K, L, M and O), while the horses were least aggressive and submissive in spring. Allogrooming is similar in all seasons. Provision of hay seems to increase agonistic behaviour (Figure 7). 

### 3.2. Group-Level Analyses 

The correlations between the numerical group characteristics (Appendix A
Appendix A) and the median frequencies of aggression on the one hand and submission on the other, for all 20 groups, follow a similar pattern (Figure 8). 

The data suggest a negative relationship between the agonistic behaviours and the number of foals present and stability, and a positive relationship with the proportion of males in the groups. The median allogrooming frequency correlated negatively with proportion of adults and group size. Considering the strong association of the presence of a stallion (which also means a low proportion of males) with agonistic interactions in this dataset (Figure 4), it was decided to remove the six groups with a stallion and undertake the same analysis with the 14 non-stallion groups (Figure 8 and Figure 9). Considering that a common practice is to have geldings, mares and sub-adults together in a group (but not with stallions), it is important to see if the correlations between the variables in Figure 8 and Figure 9 are of the same nature. When the correlation (rho’s) were strong (0.45 and higher) between a variable and one or more of the three behaviour types (Figure 8 and Figure 9) we made plots (Figure 10, Figure 11, Figure 12 and Figure 13) graphically representing the bivariate relationships.

Thus, we hope to find indications of the most important group characteristics affecting the different types of social interaction.

#### 3.2.1. Agonistic Behaviours

The strongest correlations with both of the the agonistic behaviour types were the number of foals present in the groups and stability (Figure 8 and Figure 9). In Figure 10, median aggression values for the groups are shown in relation to the number of foals in the groups.

The highest levels of aggression and submission are found in the two small sub-adult groups (A and B) and the mixed non-stallion winter groups. All the nine groups with foals present, both those with a stallion and without, show low levels of both aggression and submission (Figure 10A,B). 

In Figure 11A,B, the relations between group stability (defined in Table 3) and aggression and submission are shown for all the 20 groups.

Here the six groups with a stallion (triangle-shaped dots) are clearly different from the others, because there is no correlation between stability and median aggression nor median submission among those groups. On the other hand, there is clear negative correlation between the same variables among the 14 groups without a stallion.

Proportion of males in the groups was highly positively correlated with aggression and submission in the 20 groups (Figure 8). However, this was not found to be true for the groups without a stallion (Figure 9). This relationship is shown in Figure 12.

#### 3.2.2. Allogrooming

The strongest correlations between the group characteristics and this affiliative behaviour are all negative. Thus, allogrooming decreases with group size (Figure 13A), with higher proportion of adults (Figure 13B), with increased stability of the group (Figure 13C), and with the median number of friends (Figure 13D).

## 4. Discussion

### 4.1. Main Findings

The combined data both support earlier findings and provide new insight about effects of group composition, group stability and key environmental factors on frequencies of both agonistic and affiliative interactions. The low frequencies of agonistic interactions in the groups with a stallion present, compared to the other groups, is an interesting finding and gives a strong support to our earlier results which were based on limited data [9,23]. The presence of foals in groups correlates with low aggression. Stability of group membership is also of significance as more stability is associated with lower aggression. Allogrooming frequencies correlated negatively with two group attributes; group size and the proportion of adults. Aggression increases with provision of hay in the wintertime. 

### 4.2. Agonistic Behaviours 

The individual frequencies of aggressive interactions in these 20 groups of Icelandic horses are generally low (mean 0.42) compared to other domestic horses (means for four populations 0.9, 2.5, 3.2 and 6.3, in [25]) but similar values have been reported in some natural and semi-feral populations [25]. The frequencies of aggression do not to vary with age except for the very oldest horses, which show elevated frequencies of aggression. Interestingly, the frequency of submission is also high among the oldest horses. It appears that while they strive to keep their status by elevating aggression they run a higher risk of eliciting aggression of high-ranking individuals, causing them to submit. As has been shown repeatedly [21], submission frequencies are higher among young horses.

Clearly, the presence of stallions has a profound influence on the levels of agonistic behaviour. This agrees with results from research on feral bands with stallions, where agonistic behaviours are relatively infrequent [26]. Presence of a stallion and herding behaviour, but not direct interference, might explain the lower frequency of agonistic interactions between members within a group [23]. Also, the herding behaviour of the stallion could have some effect on the social structure and lower the level of interactions. The six stallions in this study were the most aggressive individuals of their groups (frequencies of aggression ranging from 0.11 to 0.28/h) but compared to the other horses they were relatively non-aggressive. The finding agrees with other studies where it has been shown that stallions are neither especially dominant over females or more aggressive than other horses [42,43]. Also, in pure stallion groups, aggression levels have been found to be low [44]. 

The presence and number of young foals in the groups correlated strongly and negatively with levels of aggression. This applies both to the groups with a stallion and the non-stallion groups that had foals (Figure 8, Figure 9 and Figure 10). This is a very interesting finding which to our knowledge has not been reported before. Two of those groups (D and F) were large, with pregnant and barren mares, geldings and sub-adults in addition to the dams and their foals. The two groups were studied for six weeks in the spring, during which time the number of foals gradually increased. The dams formed temporary groups after having had their foals, and defended their foals against other horses and their aggression level increased after foaling [35,36]. 

In non-stallion groups, stability of group membership has a strong overall correlation with less aggression. This is not the case among the groups with a stallion. The reason for this is not known. The presence of a stallion, the presence of foals or both factors might mask the effect of social uncertainty in the two unstable stallion groups (P and Q) (S1). In stable groups, horses are more knowledgeable about the social network and hence there is less need for the dominant horses to give strong aggressive signals [13,20,26,45]. The arrival of newcomers is associated with an increase of social interactions, most of all aggressive threats [24,25]. Prior studies have shown that there is a rapid decrease in these interactions if the membership remains unchanged [13,35]. Keiper and Sambraus pointed out, more than 30 years ago, that changes in group composition could disrupt the social organization [46]. The present study supports this and their advice to horse owners, to keep groups as stable as possible, is a very important one. The two yearling groups (A and B) had the lowest stability score and showed the highest levels of agonistic behaviours. All nine yearlings in both groups were owned by different people and were unfamiliar to each other at the beginning of the experiment. Presumably, the horses experienced very high level of stress, a situation eliciting frequent acts of aggression and submission [37]. In all the other groups, some or all of the horses were familiar to each other at the beginning of the observation. 

As might be expected, provision of hay increases agonistic interactions in most cases (an exception being the two mixed non-stallion group with foals (D and F). In Iceland, the most common practice is to deliver hay in large rolls that are placed in the pasture without distributing the contents any further. Delivering the hay in this way means that the horses are in close proximity to each other while eating. This seems to increase aggression levels (personal observations). For these reasons we recommend that farmers and other caretakers should ensure that the subordinate horses get good access to supplementary hay. 

The season does appear to have some influence on the frequencies of social interactions. The fact that agonistic behaviours increase in the wintertime is probably both because of the effect of the method of delivering supplementary hay and diminished opportunities for grazing. 

### 4.3. Affiliative Behaviour

The findings that allogrooming frequencies correlated strongly and negatively with group size and the proportion of adults in groups has to our knowledge not be published before. The reason for the former could be that group size is a limiting factor for a good social network to develop, where all individuals have opportunities to interact with each other [31]. The reasons for the latter finding, i.e., that the horses allogroom less in groups where adults are relatively numerous, is not known, but the following explanations are plausible. The older horses have already formed bonds while that is not the case with the younger ones and since the frequency of affiliative interactions decreases when social bonds are established [3,13] such a correlation is to be expected. The finding that younger horses allogroom more than the older ones [this study], agrees with this. 

Allogrooming was also negatively correlated with group stability. Horses in unstable groups are likely to experience increased stress levels, because of high frequencies of agonistic interactions [this study]. In stressful situations, the reaction to increase mutual grooming can have an adaptive value because of the positive emotions and lowered blood pressure that the horses are probably experiencing [7]. At the same time, the need to form and maintain social bonds in unstable groups is present. However, as that need diminishes with increased familiarity (stability), the horses show a preference to allogroom with certain individuals (friends). Interestingly, the horses allogroom less when they have more friends (Figure 8, Figure 9 and Figure 13). The fact that group size and median number of friends is positively correlated (S1, rho = 0. 46) and that allogrooming frequencies are lower in large groups supports the notion that when horses have formed strong bonds with one or two individuals, they show extra high level of affiliation towards these individuals. 

## 5. Conclusions

This study emphasizes the need for horse caretakers to consider group stability and group composition in domestic horse management. This study is based on observations of horses held in spacious pastures but the findings also apply to the organization of group housing and the use of paddocks. Group composition which is similar to the natural social system with both sexes (sub-adults and foals) and stable membership of adult horses is likely to provide the best social environment for all horses because of very low aggression levels. Also, it is likely to offer the best conditions for young horses to learn social skills, because of the presence of older and experienced horses and the possibilities to associate with peers Such groups, or groups where stallions are replaced by adult geldings, could easily be taken up as a management practice. Peer groups should be avoided, especially if composed of young unfamiliar horses, because of high levels of aggression. 

The level of allogrooming was more similar in the different group types than the frequency of agonistic behaviours. The two small unstable yearling groups had the highest allogrooming levels. The horses allogroomed less in large groups where they had more friends, but interestingly the horses that had few friends, allogroomed most.

Although only descriptive statistics were applied because of the nature of the data, the results of this paper are very interesting and should, first, provide a strong message to horse owners and caretakers and, second, encourage further research to test which factors are of the greatest significance. The importance of the welfare of horses can never be overemphasized. Recent research on the cognitive abilities of horses and their emotions [47] has shown that horses are more complex in these areas than previously thought. It is our belief that awareness of such findings will affect both personal views and legislation on horse welfare in the near future.

## Figures and Tables

**Figure 1 animals-09-00014-f001:**
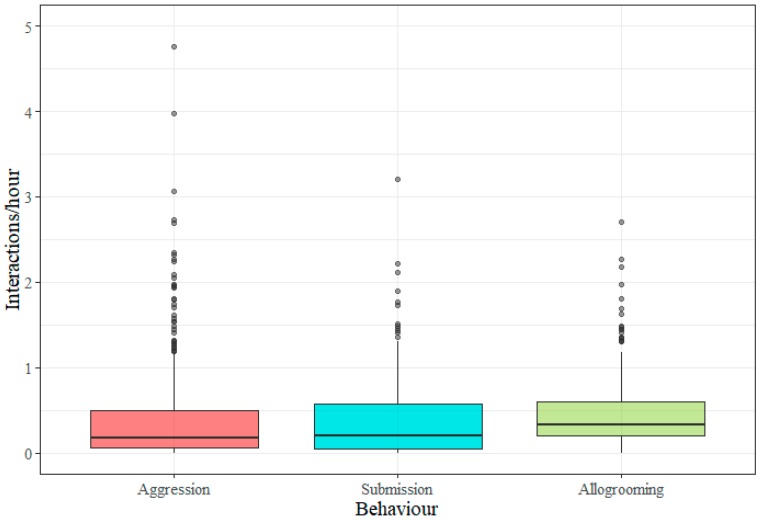
Boxplots showing the distribution of individual social interaction frequencies (per hour) for aggression, submission and allogrooming. Data from 426 horses are displayed. Medians for aggression and submission were 0.18/h and 0.20/h, respectively, and the corresponding averages were 0.42/h (standard error (SE) = 0.032) and 0.35/h (SE = 0.020). The median frequency of allogrooming across groups was 0.33/h and the mean was 0.43/h (SE = 0.018). The aggression frequency from one outlier was excluded (see Section 2.4).

**Figure 2 animals-09-00014-f002:**
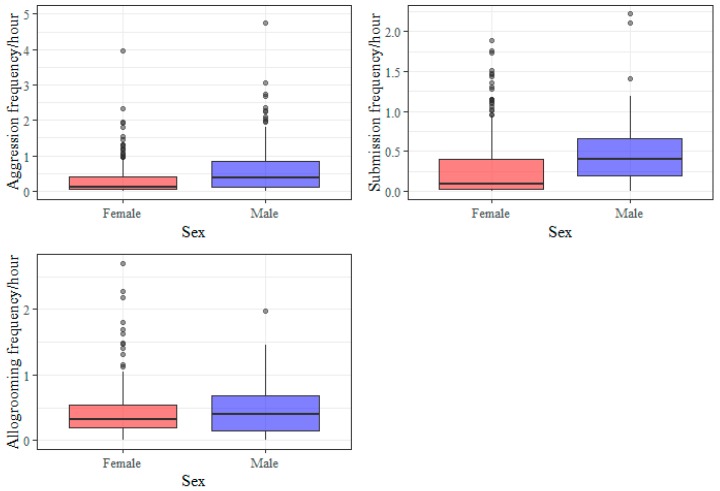
Boxplots showing the distribution of individual social interaction frequencies (per hour) for aggression, submission and allogrooming by sex. Data from 297 female horses and 129 male horses (118 geldings, six stallions, five colts) are displayed. Median aggression rate for the stallions was 0.54 (IQR = 0.36). Median submission rate was 0.20 (IQR = 0.11). Median allogrooming rate was 0.20 (IQR = 0.20). The aggression frequency from one outlier was excluded (see Section 2.4).

**Figure 3 animals-09-00014-f003:**
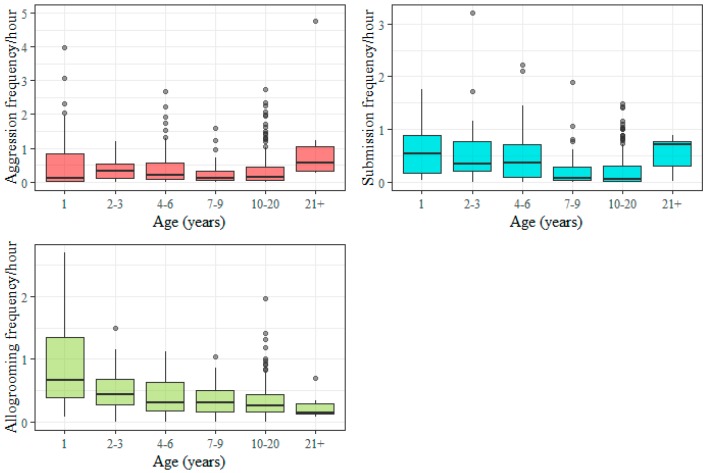
Boxplots showing the distribution of individual social interaction frequencies (per hour) for aggression, submission and allogrooming by age class. For some groups exact age was known for all or majority of the animals but because information was limited for other groups (I, R, S, T, U) broad age classes had to be used for mature horses. Information on 20 horses was not available. The number of individuals in each age class are: A1: *n* = 44, A2: *n* = 91, A3: *n* = 58, A4: *n* = 50, A5: *n* = 156, A6: *n* = 7. The aggression frequency from one outlier was excluded (see Section 2.4).

**Figure 4 animals-09-00014-f004:**
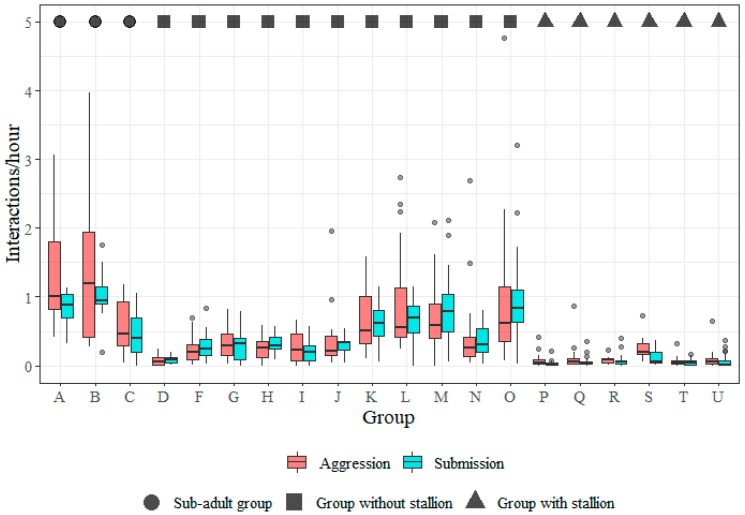
Boxplots showing the distribution of individual social interaction frequencies (per hour) for aggression and submission by group. Description of groups are given in Table 1. The aggression frequency from one outlier was excluded (see Section 2.4).

**Figure 5 animals-09-00014-f005:**
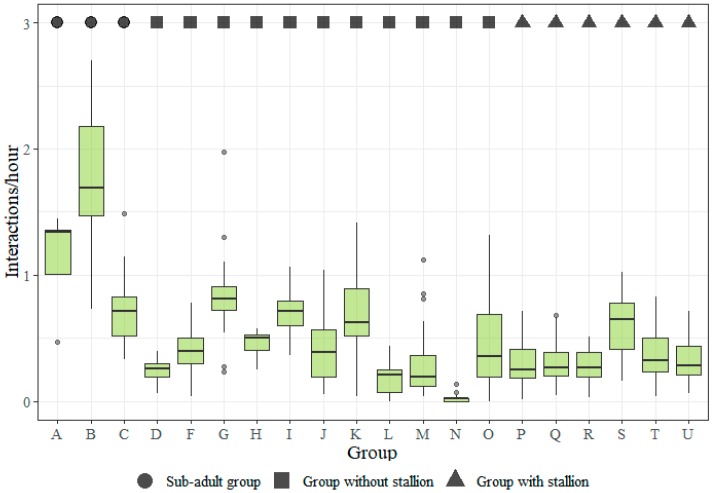
Boxplots showing the distribution of individual social interaction frequencies (per hour) for allogrooming by group. Description of groups are given in Table 1.

**Figure 6 animals-09-00014-f006:**
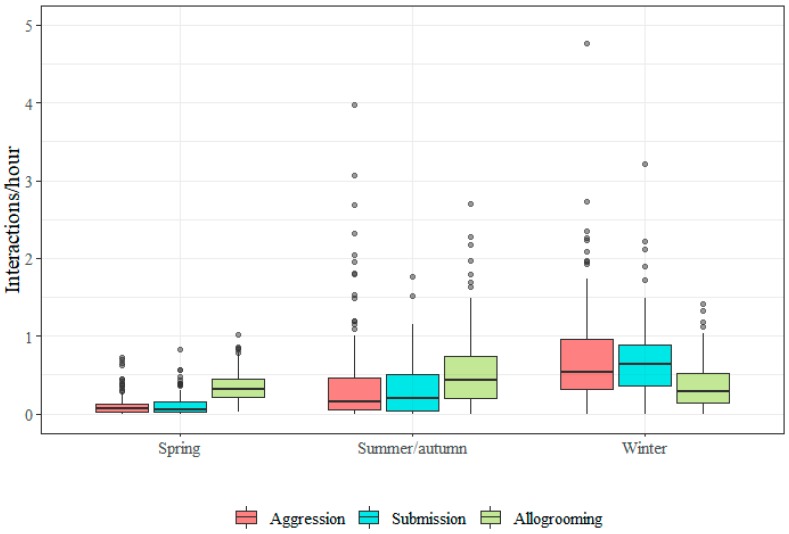
Boxplots showing the distribution of individual social interaction frequencies (per hour) for aggression, submission and allogrooming by season. Data from 20 horses observed in autumn, 144 in spring, 137 in summer, and 125 in winter are displayed. Only one group was observed in autumn and their data are combined with the data from the summer groups. The aggression frequency from one outlier was excluded (see Section 2.4).

**Figure 7 animals-09-00014-f007:**
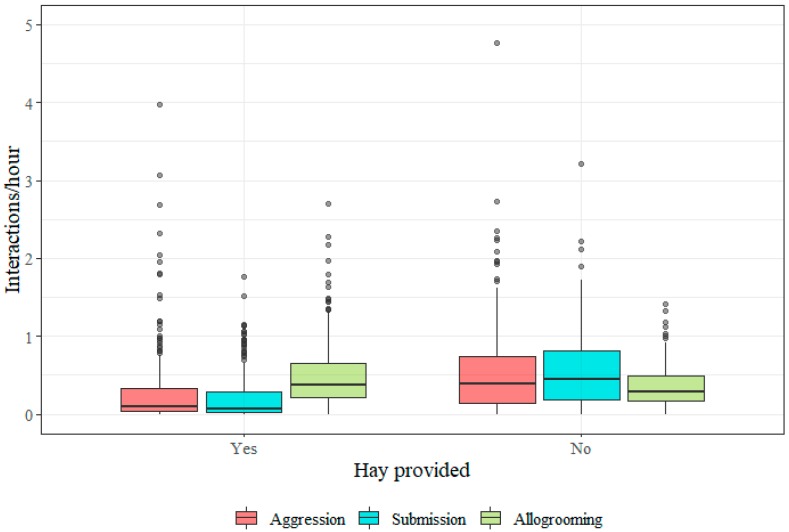
Boxplots showing the distribution of individual social interaction frequencies (per hour) for aggression, submission and allogrooming by provision of hay. Data from 176 horses receiving hay and 250 not receiving hay is displayed. The aggression frequency from one outlier was excluded (see Section 2.4).

**Figure 8 animals-09-00014-f008:**
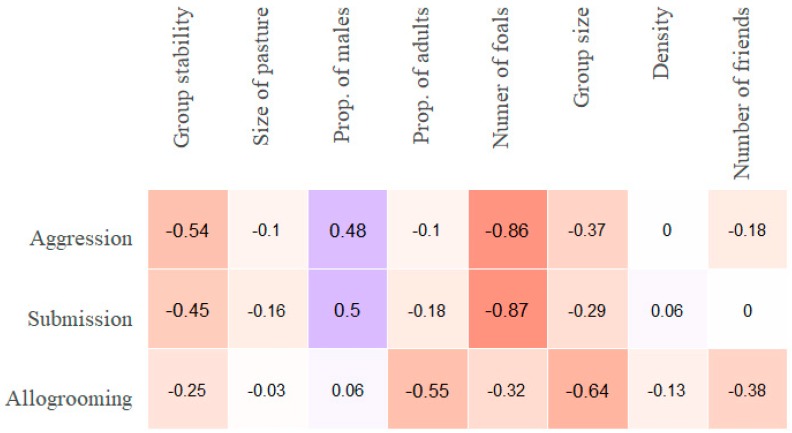
Spearman rank-order correlations (rho) of all group-level variables with group medians of individual frequencies (per hour) of aggression, submission and allogrooming for the 20 groups. The colour coding is employed to facilitate the visualization of the strength of correlations. “Number of friends” refers to the median number of friends (allogrooming associates) in the group.

**Figure 9 animals-09-00014-f009:**
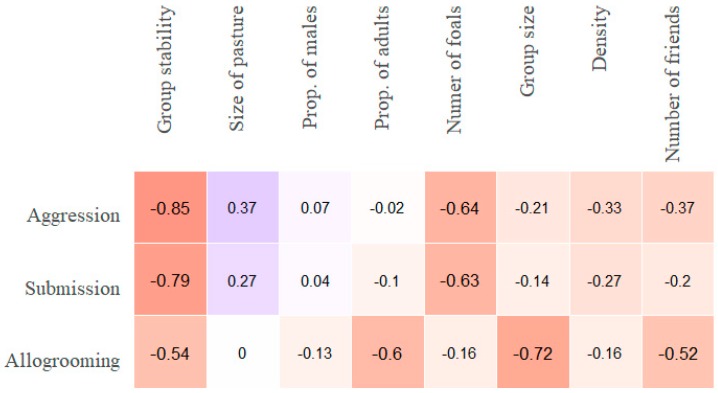
Spearman rank-order correlations (rho) of all group-level variables with group median individual aggression, submission and allogrooming frequencies (per hour) for the 14 non-stallion groups. The colour coding is employed to facilitate the visualization of the strength of correlations. “Number of friends” refers to the median number of friends (allogrooming associates) in the group.

**Figure 10 animals-09-00014-f010:**
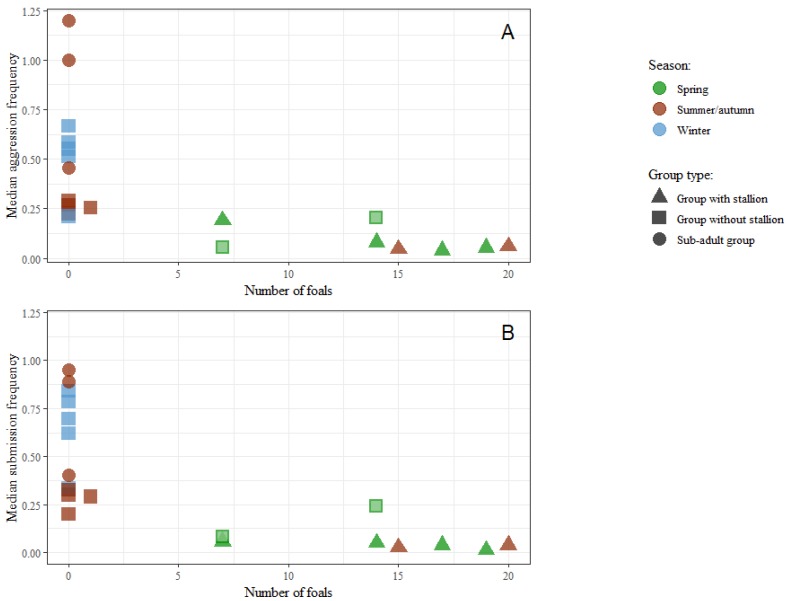
Scatterplots of group medians for both individual aggression (**A**) and individual submission (**B**) frequencies by number of foals in a group. All groups observed in winter were provided with hay. Two groups observed in spring were also provided with hay and are marked with a light centre.

**Figure 11 animals-09-00014-f011:**
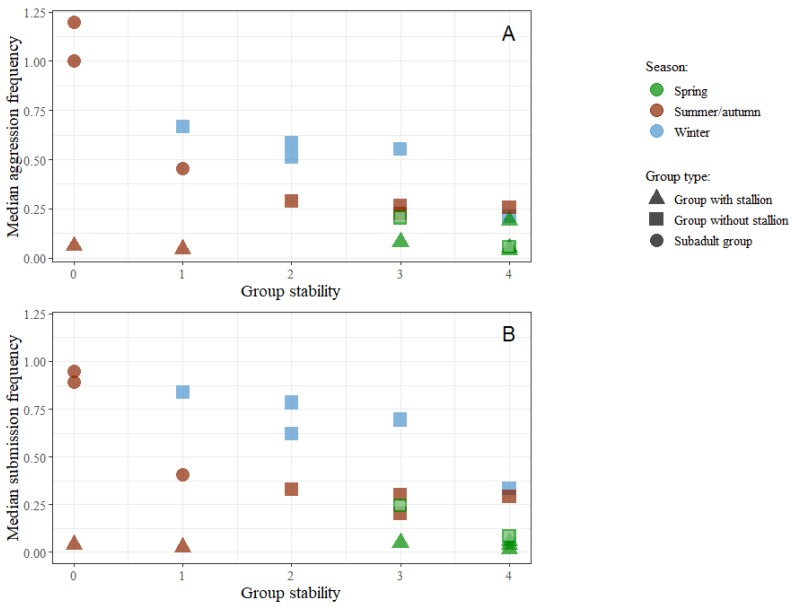
Scatterplots of group medians for both individual aggression (**A**) and submission (**B**) frequencies by stability of the groups (defined in Table 3). All groups observed in winter were provided with hay. Two groups observed in spring were also provided with hay and are marked with a light centre.

**Figure 12 animals-09-00014-f012:**
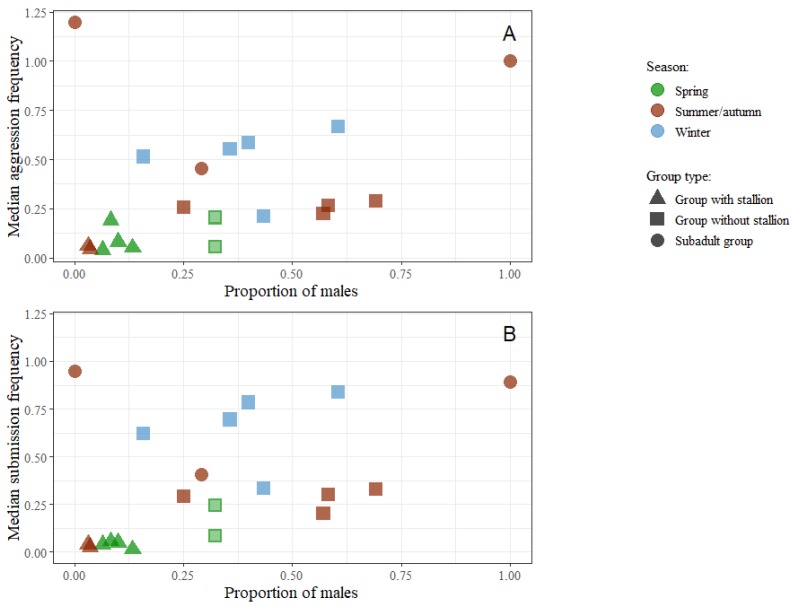
Scatterplots of group medians for both individual (**A**) aggression and submission (**B**) frequency by proportion of males. All groups observed in winter were provided with hay. Two groups observed in spring were also provided with hay and are marked with a light centre.

**Figure 13 animals-09-00014-f013:**
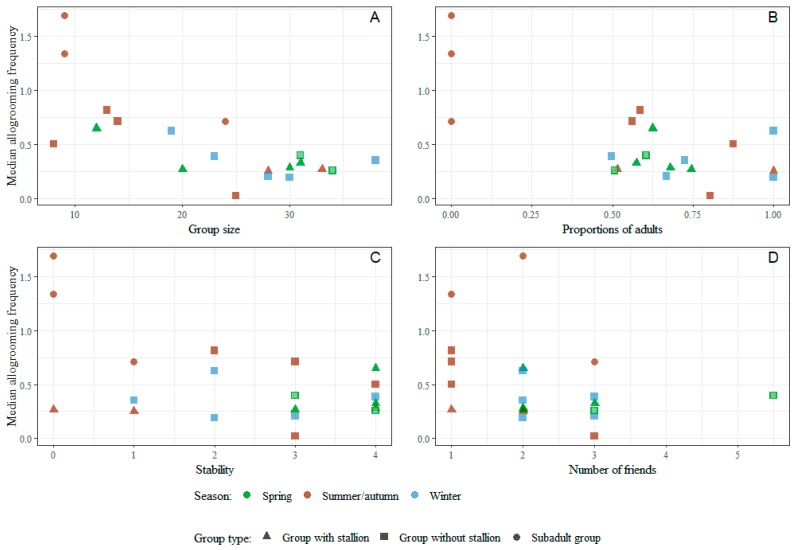
Scatterplots of group median individual allogrooming frequency (per hour) by group size (**A**), proportion of adults (**B**), stability (**C**) and median number of friends (preferred partners) (**D**). All groups observed in winter were provided with hay. Two groups observed in spring were also provided with hay and are marked with a light centre. “Number of friends” refers to the median number of friends (allogrooming associates) in the group.

**Table 1 animals-09-00014-t001:** Information on the groups.

Part of Iceland, Place, Year, Months of Study	Group ID	Females	Males *	Foals	Sub-Adults	Adults	Observation Time (hours)	Pasture Size (ha)
NW, Bessastadir—2005 (7–8)	A^37^	0	9	0	9	0	100	30
NW, Bessastadir—2005 (7–8)	B^37^	9	0	0	9	0	100	100
N, Holar—2003 (6–7)	C	17	7	0	24	0	79	5.4
W, Skaney—1997 (5–6)	D^9,35^	23	11	14	12	22	847	8
W, Skaney—1999 (5–6)	F^35^	21	10	7	6	25	488	8
SW, Litla-Thufa—2012 (7)	G	4	9	0	4	9	40	6.5
SW, Middalur—2012 (7–8)	H	6	2	1	7	1	40	30
SW, Eilifsdalur—2012 (7–8)	I	6	8	0	9	5	55	35
N, Holar—2001 (2–4)	J^33^	13	10	0	7	16	57	26.5
N, Holar—2001 (2–4)	K^33^	16	3	0	0	19	55	27.5
N, Holar—2001–2002 (12–5)	L^33^	18	10	0	20	8	102	26.5
N, Holar—2002 (1–5)	M^33^	18	12	0	0	30	81	27.5
SW, Baer—2009 (10–12)	N^38^	10	14	0	4	20	44	100
SW, Fell—2009 (1–3)	O^38^	15	23	0	6	32	41	30
* NW, Thoreyjan.—2004 (7–8)	P^23^	27	1	15	0	28	76	30
* NW, Thingeyrar—2006 (6–7)	Q^23^	32	1	20	0	33	133	8
** S, Sel 1—2007 (5)	R^23,34^	18	*** 2 (1)	14	3	17	81	215
** S, Sel 2—2007 (5)	S^23,34^	11	1	7	3	9	81	215
** S, Sel 3—2007 (5)	T^23,34^	29	*** 2 (1)	17	10	21	77	215
** S, Sel 4—2007 (5)	U^23,34^	26	*** 4 (3)	19	6	24	77	215

* The majority of individuals were geldings (*n =* 118), with *n =* 6 stallions and *n =* 5, 7–10 month old colts. ** Groups with a stallion. *** Number of colts in brackets. NW is North-West, N is North, W is West, SW is South–West, S is south. The superscripts on group IDs refer to previous published analyses of the data from the groups.

**Table 2 animals-09-00014-t002:** Types of social behaviour measured and criteria used.

Variable	Definitions/Explanations (For Detailed Description for Each Behaviour Type—See McDonell [8].)
Aggression [8]	Obvious aggression—fight, bite, threat to bite, attack, threat with ears pinned backwards, aggressive chase, strike with foreleg. All these behaviour types were recorded.
Submission—see [8]	Both subtle reactions in the form of avoidance and strong movements (running) due to aggression shown by another horse. The behaviour types recorded were: supplant (yield) by moving the head or upper body away from the aggressor, move away with leg movement and flee with fast leg movement.
Allogrooming or Mutual grooming—see [8]	Two horses stand head to tail and scratch the skin of the other horse with their teeth. When horses stopped allogrooming and started again, one minute had to elapse between the sessions to start counting a new session.

**Table 3 animals-09-00014-t003:** Definitions of ranked variables which are used in the analyses.

**Age classes.** A1: 1 year, A2: 2–3 years, A3: 4–6 years, A4: 7–9 years, A5: 10–20 years, A6: 21 and more. Twenty adult horses could not be assigned to a class. In Iceland a year is added to the age of a horse on the first day of summer in early April.
**Season.** autumn (October–December), winter (January–April), spring (May–June), summer (July–August).
**Stability.** 0: >5 owners, 1: 4–5 owners, 2: 2–3 owners, 3: 1 owner. The classification assumes that the number of owners/caretakers of the horses in a group reflects the probability of changes in the composition (permanent and short term removals and introductions). If all horses were unfamiliar in the beginning of the observation period they got the lowest score (0).
**Stallions.** Absence in a group, (0), presence in a group, (1).
**Number of associates (friends).** The number of horses with which a horse allogrooms significantly more often than if he allogroomed with all in his group in a random manner (Chi-squared analyses, *p* < 0.05). If stay times in the groups differed between horses, the predicted values for the relevant dyads were corrected accordingly.

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
