# Peer review of "Significance of Group Composition for the Welfare of Pastured Horses"

_animals, 2019, doi:10.3390/ani9010014_

Round 1
Reviewer 1 Report
It is good to see a meta-analysis approach being taken to the examination of social relations in horses. However there are concerns from the outset - with an evident adherence to the outdated concept of Dominance hierarchies in horses (or most other spp for that matter). Social networks were referred to but not explained to any extent. The introduction contained quite alot of conjecture and really would benefit from a substantial rewrite with a clearer framework. It was really rather descriptive in places.
Throughout the authors included a lot of preamble at the start of sentences, resulting in quite alot of repetition throughout the paper. It is not necessary (and quite distracting) to include signposting to other parts of the report/paper if it is well written. I have highlighted several occurrences for removal to improve the flow.
In the Method/Results there is alot of justification of approaches that were NOT used, this is very distracting and undermines the methods that were used.
PLEASE remember that the word data = plural, so data are/were throughout.
The figures were useful but a little arduous to get through. If there was a stronger/clearer framework for the entire paper it would be easier to work from them.
Please avoid 'seemed to' when describing results - this completely undermines any statistical (albeit descriptive) analysis undertaken.
The discussion, although divided into sections, failed to give the 'end of the story'. There were several cases of colloquial wording and the arguments following the analysis to answer the paper question/title never quite developed.
The conclusion requires a rewrite - in order to align with the title and to be sufficiently succinct with a clear take-home message.
I have made several (n=113) comments on the submitted manuscript (supplied separately).

Author Response
It is good to see a meta-analysis approach being taken to the examination of social relations in horses. However there are concerns from the outset - with an evident adherence to the outdated concept of Dominance hierarchies in horses (or most other spp for that matter). Social networks were referred to but not explained to any extent. The introduction contained quite alot of conjecture and really would benefit from a substantial rewrite with a clearer framework. It was really rather descriptive in places.
Response : We have reacted to these points: 1.The dominance hierarchy concept is not used anymore. 2. Social networks were replaced by social bonding and the text simplified, 3. The introduction has been shorten and improved in light of these comments
Throughout the authors included a lot of preamble at the start of sentences, resulting in quite alot of repetition throughout the paper. It is not necessary (and quite distracting) to include signposting to other parts of the report/paper if it is well written. I have highlighted several occurrences for removal to improve the flow.
Response: The manuscript is shorter (by ca. 1000 words) and clearer, thanks to good comments by you and the other reviewers. We have tried to improve the style by rewording and shortening sentences.
In the Method/Results there is alot of justification of approaches that were NOT used, this is very distracting and undermines the methods that were used.
Response: The statistics section has been shortened and is much more focused. We deleted repetitions and moved some of the sentences to the discussion chapter. moved
PLEASE remember that the word data = plural, so data are/were throughout. .
Response: We did.
The figures were useful but a little arduous to get through. If there was a stronger/clearer framework for the entire paper it would be easier to work from them.
Response: As the whole ms is more focused now we hope this is no longer a problem.
Please avoid 'seemed to' when describing results - this completely undermines any statistical (albeit descriptive) analysis undertaken.
Response: done.
The discussion, although divided into sections, failed to give the 'end of the story'. There were several cases of colloquial wording and the arguments following the analysis to answer the paper question/title never quite developed.
The conclusion requires a rewrite - in order to align with the title and to be sufficiently succinct with a clear take-home message.
Response: The discussion and conclusions have been rewritten and shortened ( by ca 200 words) and are more focused now with a clear message.
We can not read comment number 75 (on Figure 4) and do not understand your comments number 56 and 57.
I have made several (n=113) comments on the submitted manuscript (supplied separately). Thank you.
Reviewer 2 Report
See attached paper.
This report contains findings on behavior of individual horses within different groups and management factors. The information has been published previously but in this report the studies are combined to highlight findings over a large number of studied animals and groups. I find the topic of large interest with the change across the world to more group housing and recognizing the need for group housing of horses. In general the report is well written and the graphs are clear. My overall concern is that the manuscript is too long and the main conclusions are buried within the report. It is hard for a reader to gain the most important facts from this paper. I suggest shortening and writing the report to highlight the conclusions (for example one main conclusion is that horses are best housed in a group with different ages and genders, which is contrary to what most barns do?). My specific suggestions are listed here.
Summary:
Line 13: I don’t think you need to include (14)
Abstract:
Line 20: potentially important factors??? Do you mean heard composition /management factors?
Line 26: group type?? Not clear what is meant by that if you don’t read the paper first.
Line 28 – group stability – this is also a term that needs more explanation before using it. So I would not include it in the abstract but instead say something like groups with low frequency of horse movement? Stability can refer to composition of individual horses within a group (horse turn over) or stability of 1 horses that is possibly moved around from one owner to the next to the next (possibly big effect on individual horse and small effect on remainder of herd?).
Introduction – way too long
Line 42 – good welfare – based on what? Perhaps including a reference that shows that horses have lower incidences of medical problems and lower heart rate / stereotypic behaviors when housed together?
Line 60 – I would expand on what is considered stable. What is group stability? Section of line 86 might move forward.
Line 105 – remove see below. Remove the section from Line 106 – 114 = M&M
Following line 105 continue with line 124 (modified to make read better; ex. By combining data from 20 studies that have been previously published….). Remove Line 126-128. Then continue with Line 115. Summarize Lines 115-123 into 1 sentence.
M&M –
2.1 line 136 – the country = Iceland.
Table 1 – what about geldings vs. stallions; I think they should be separated out? Also, is there anything known on what the effect is of having geldings and stallions in a group? (that would not be a natural thing so comparing to natural environment difficult?)
The Table should be listed under 2.1 not 2.2
2.2
Should Table 2 be referenced as coming from [7]? Reference behind title and include Taken from or similar.
Line 185 – is there a reference for 90% agreement? If not, I think that paragraph needs to be removed or worded differently.
2.4
Line209-238 – check with statistician possibly is discussion of limitations and should be moved to the discussion section. Here just state what you did and not what and why you did something else.
Line 253-257 - discussion
Line 251 – Is there a particular reason to remove this horse’s data? Was the horse ill? I wonder whether the horse’s data should be included. Reason for removal needs to be stated.
Results
Line 265 – Figure 1 the word “chapter” is mentioned – this is a very long manuscript indeed, but for this that should probably be changed to section.
Line 266 – few individuals. In Figures 1 are the outliers (positive skewed data) the same individuals for the 3 types of behavior? Are the outliers the same individuals across Figures 1 – 3?
Line 272-276: delete
Figure 2 – what about gelding vs. mare vs stallion?
Line 319-322 – delete from here and include in discussion
Figure 8 and 9 highlight some effects of having stallion in stead of geldings in a group. That is good information.
Figure 11 – the term stability needs to be explained.
Discussion
Lines 406-407 – what is the frequency in other horses? Reference?
Overall the results/discussion/conclusion contain much duplication and could all be shortened up. I suggest picking the important points the research shows and presenting those as paragraphs – finding, discussion and conclusion. It now reads a bit like a hodg-pot of information and that might make the reading and drawing of those important conculsions easier for the reader.

Author Response
This report contains findings on behavior of individual horses within different groups and management factors. The information has been published previously but in this report the studies are combined to highlight findings over a large number of studied animals and groups. I find the topic of large interest with the change across the world to more group housing and recognizing the need for group housing of horses. In general the report is well written and the graphs are clear. My overall concern is that the manuscript is too long and the main conclusions are buried within the report. It is hard for a reader to gain the most important facts from this paper. I suggest shortening and writing the report to highlight the conclusions (for example one main conclusion is that horses are best housed in a group with different ages and genders, which is contrary to what most barns do?). My specific suggestions are listed here.
Response: 1. Most but not all the information in the manuscript has been published before. 2. As you know we focused on pastured horses but the results are useful for horseowners keeping horses in group houses and in spacious paddocs as well. 3. The manuscript has shortened considerably (by ca 1000 words- Introduction by 300 and Discussion and conclusion by 200 words . 4. The main conclusions are drawn out much clearer now and the structure of the last part (D and C) has changed in lightof your comments and those of other reviewers.
Summary: All done
Line 13: I don’t think you need to include (14)
Abstract:- All done
Line 20: potentially important factors??? Do you mean heard composition /management factors?
Line 26: group type?? Not clear what is meant by that if you don’t read the paper first.
Line 28 – group stability – this is also a term that needs more explanation before using it. So I would not include it in the abstract but instead say something like groups with low frequency of horse movement? Stability can refer to composition of individual horses within a group (horse turn over) or stability of 1 horses that is possibly moved around from one owner to the next to the next (possibly big effect on individual horse and small effect on remainder of herd?).
Introduction – way too long. Has been shortened.
Line 42 – good welfare – based on what? Perhaps including a reference that shows that horses have lower incidences of medical problems and lower heart rate / stereotypic behaviors when housed together?. Done
Line 60 – I would expand on what is considered stable. What is group stability? Section of line 86 might move forward. Done
Line 105 – remove see below. Remove the section from Line 106 – 114 = M&M Done
Following line 105 continue with line 124 (modified to make read better; ex. By combining data from 20 studies that have been previously published….). Remove Line 126-128. Then continue with Line 115. Summarize Lines 115-123 into 1 sentence. We have made revisions in light of this.
M&M –
2.1 line 136 – the country = Iceland. Done
Table 1 – what about geldings vs. stallions; I think they should be separated out? Also, is there anything known on what the effect is of having geldings and stallions in a group? (that would not be a natural thing so comparing to natural environment difficult?) the table has been revised. In Iceland we rarely keep stallions and geldings together and no such group was in the dataset.
The Table should be listed under 2.1 not 2.2 Done
2.2
Should Table 2 be referenced as coming from [7/8 now].
Reference behind title and include Taken from or similar. Done
Line 185 – is there a reference for 90% agreement? Yes If not, I think that paragraph needs to be removed or worded differently.
2.4
Line209-238 – check with statistician possibly is discussion of limitations and should be moved to the discussion section. Here just state what you did and not what and why you did something else. Done
Line 253-257 – discussion. Done
Line 251 – Is there a particular reason to remove this horse’s data? Was the horse ill? I wonder whether the horse’s data should be included. Reason for removal needs to be stated. Done
Results
Line 265 – Figure 1 the word “chapter” is mentioned – this is a very long manuscript indeed, but for this that should probably be changed to section. Done
Line 266 – few individuals. In Figures 1 are the outliers (positive skewed data) the same individuals for the 3 types of behavior? Are the outliers the same individuals across Figures 1 – 3? This information has been added. Done
Line 272-276: delete. Done
Figure 2 – what about gelding vs. mare vs stallion? Information in the figure text has been added.
Line 319-322 – delete from here and include in discussion. Done
Figure 8 and 9 highlight some effects of having stallion in stead of geldings in a group. That is good information.
Figure 11 – the term stability needs to be explained. Done
Discussion
Lines 406-407 – what is the frequency in other horses? Reference? Done, although we are sceptical of having means in the text (rather than medians), but it was necessary to be able to compare with publshed results (Fureix et al)
Overall the results/discussion/conclusion contain much duplication and could all be shortened up. I suggest picking the important points the research shows and presenting those as paragraphs – finding, discussion and conclusion. It now reads a bit like a hodg-pot of information and that might make the reading and drawing of those important conculsions easier for the reader. We have shortened the discussions and followed your advice.
Reviewer 3 Report
I really enjoyed reading this paper and think the findings have significance for horse management practices.
It is well written on the whole, and just requires some minor proof reading for grammar by a native English speaker.
It is based on a very strong data set and although results have been published previously, this paper draws multiple studies together and does contribute something new.
I have only a few very small comments.
The simple summary is not as informative as the abstract. Although the task is to keep it simple, I think information about things like number of horses in total are easy to understand and relevant to understanding the strength of the data on which the paper is based.
It is a little confusing throughout in terms of what is meant by 'male horses'. Is this only stallions, or stallions and geldings? Given that geldings are far more prevalent in the leisure market, and therefore welfare issues related to their management keep very important, this needs to be clarified and the findings in relation to geldings could be made clearer.
Conclusions - I think the authors could usefully point to some of the practical implications of the findings in terms of horse management and how this reflects or not current practices as this has potential to affect welfare considerably.
Author Response
I really enjoyed reading this paper and think the findings have significance for horse management practices.
It is well written on the whole, and just requires some minor proof reading for grammar by a native English speaker.
Response: Done.
It is based on a very strong data set and although results have been published previously, this paper draws multiple studies together and does contribute something new.
I have only a few very small comments.
The simple summary is not as informative as the abstract. Although the task is to keep it simple, I think information about things like number of horses in total are easy to understand and relevant to understanding the strength of the data on which the paper is based.
Response: We have added information.
It is a little confusing throughout in terms of what is meant by 'male horses'. Is this only stallions, or stallions and geldings? Given that geldings are far more prevalent in the leisure market, and therefore welfare issues related to their management keep very important, this needs to be clarified and the findings in relation to geldings could be made clearer.
Response: We have added information to table 1 and the figure text for figure 2 and some text to the results section that should address this.
Conclusions - I think the authors could usefully point to some of the practical implications of the findings in terms of horse management and how this reflects or not current practices as this has potential to affect welfare considerably.
Response: The discussion and conclusions have been substantially revised in light of other reviewers comments and is shorter and more focused now. We address horse owners in these sections and encourage them to adapt their management practice.
Thank you for your comments.
Round 2
Reviewer 1 Report
Thank you for reviewing this paper so promptly. I really appreciate the attention to detail you have displayed and the comprehensive approach taken to producing Revision 1. I thoroughly enjoyed undertaking the review of R1.
I have a few additional, albeit very minor, changes/edits: [ONLY N=12 THIS TIME!).
Under Table 1. Change *Most are geldings, stallions were six in total, 7-10 months old colts were five to ‘*the majority of individuals were geldings (n=XX), with n=6 stallions and n=5 7-10 month old colts’
Table 3. Replace ‘friends’ with ‘associate’.
Line 226. ‘Boxplots are presented in the standard Tukey style” – I am not aware that this is the standard tukey style and you don't really need to say so. Remove ‘Boxplots are presented in the standard Tukey style’.
Line 256/257. The text on line 257 can follow directly on after the text that ends on line 256.
Figure 3. x-axis, age class, requires units (years).
Figure 6. Addition to legend useful, can you just change: ‘Only one group was observed in autumn and 299 their data is combined with the data from the summer groups.’ To ‘Only one group was observed in autumn and 299 their data ARE J combined with the data from the summer groups.
Figures 8 & 9. Colour is OK if printed in colour, need to be mindful of colour blind readers. Also needs a description included in the legend. Can you do something like bold font, bold italic font and italic font for P<0.05; P<0.01 and P<0.001 correlations, and plain for non sig P= or > 0.05 correlations?
Lines 391/392 Line 392 could continue on from line 391 making this subsection a single paragraph.
Line 397. Re. ‘Provision of hay in the wintertime increases with aggression.’ – did you mean it this way around?
Line 429. Remove line space.
Line 446. Re. new text ‘in most cases (exception…’ insert AN or THE between ‘cases’ and ‘(exception)
OOPS – apologies that my original set of comments had pages 21 and 22 in reverse order.
Line 495. Suggest changing (minorly) ‘we stress that the results are very interesting and should both give a strong message to horse owners and caretakers and encourage further research to test which factors are of the greatest significance’ to ‘the results of this paper are very interesting and should first, provide a strong message to horse owners and caretakers and second, encourage further research to test which factors are of the greatest significance…
Author Response
Thank you for reviewing this paper so promptly. I really appreciate the attention to detail you have displayed and the comprehensive approach taken to producing Revision 1. I thoroughly enjoyed undertaking the review of R1.
Thank you.
I have a few additional, albeit very minor, changes/edits: [ONLY N=12 THIS TIME!).
Under Table 1. Change *Most are geldings, stallions were six in total, 7-10 months old colts were five to ‘*the majority of individuals were geldings (n=118), with n=6 stallions and n=5,7-10 month old colts’
Done
Table 3. Replace ‘friends’ with ‘associate’Line 226. ‘Boxplots are presented in the standard Tukey style” – I am not aware that this is the standard tukey style and you don't really need to say so. Remove ‘Boxplots are presented in the standard Tukey style’.
Done We added friends in brackets in llight of our arguments in lines 46-52.
Line 256/257. The text on line 257 can follow directly on after the text that ends on line 256.
Done.
Figure 3. x-axis, age class, requires units (years).
Done.
Figure 6. Addition to legend useful, can you just change: ‘Only one group was observed in autumn and 299 their data is combined with the data from the summer groups.’ To ‘Only one group was observed in autumn and 299 their data ARE Jcombined with the data from the summer groups.
Done
Figures 8 & 9. Colour is OK if printed in colour, need to be mindful of colour blind readers. Also needs a description included in the legend. Can you do something like bold font, bold italic font and italic font for P<0.05; P<0.01 and P<0.001 correlations, and plain for non sig P= or > 0.05 correlations?
The groups are ultimately a convenience sample and in a strict sense valid p-values cannot be obtained. While p-values could in this case perhaps be used as a measure of effect size we are trying to reform our use of p-values in line with the 2016 ASA statement on p-values, which advises against this practice (see hyperlink):
https://amstat.tandfonline.com/doi/full/10.1080/00031305.2016.1154108?scroll=top&needAccess=true#.XCI3s1z7RPY
We have made the colours more transparent to increase the contrast with the letters and help colour blind readers read the text. We have also made the letters for correlations over 0.45 (which we plot) slightly larger to make them more salient. We did experiment with using font size to represent the strength of correlations but this was not successful.
Lines 391/392 Line 392 could continue on from line 391 making this subsection a single paragraph.
Done.
Line 397. Re. ‘Provision of hay in the wintertime increases with aggression.’ – did you mean it this way around?
Done – thanks for catching this error.
Line 429. Remove line space.
Done.
Line 446. Re. new text ‘in most cases (exception…’ insert AN or THE between ‘cases’ and ‘(exception)
Done.
OOPS – apologies that my original set of comments had pages 21 and 22 in reverse order.
Line 495. Suggest changing (minorly) ‘we stress that the results are very interesting and should both give a strong message to horse owners and caretakers and encourage further research to test which factors are of the greatest significance’ to ‘the results of this paper are very interesting and should first, provide a strong message to horse owners and caretakers and second, encourage further research to test which factors are of the greatest significance…
Done.
We have also added one sentence to the figure texts for figures 8, 9 and 13 to clarify what “Number of friends” refers to in these figures.
Thank you.